# Structural Insights into Common and Host-Specific Receptor-Binding Mechanisms in Algal Picorna-like Viruses

**DOI:** 10.3390/v14112369

**Published:** 2022-10-27

**Authors:** Han Wang, Anna Munke, Siqi Li, Yuji Tomaru, Kenta Okamoto

**Affiliations:** 1The Laboratory of Molecular Biophysics, Department of Cell and Molecular Biology, Uppsala University, 75124 Uppsala, Sweden; 2Center for Free-Electron Laser Science CFEL, Deutsches Elektronen-Synchrotron DESY, 22607 Hamburg, Germany; 3Fisheries Technology Institute, Japan Fisheries Research and Education Agency, Hatsukaichi 739-0452, Hiroshima, Japan

**Keywords:** *Marnaviridae*, algal bloom, algal viruses, icosahedral viruses, ssRNA viruses

## Abstract

*Marnaviridae* viruses are abundant algal viruses that regulate the dynamics of algal blooms in aquatic environments. They employ a narrow host range because they merely lyse their algal host species. This host-specific lysis is thought to correspond to the unique receptor-binding mechanism of the *Marnaviridae* viruses. Here, we present the atomic structures of the full and empty capsids of Chaetoceros socialis forma radians RNA virus 1 built-in 3.0 Å and 3.1 Å cryo-electron microscopy maps. The empty capsid structure and the structural variability provide insights into its assembly and uncoating intermediates. In conjunction with the previously reported atomic model of the Chaetoceros tenuissimus RNA virus type II capsid, we have identified the common and diverse structural features of the VP1 surface between the *Marnaviridae* viruses. We have also tested the potential usage of AlphaFold2 for structural prediction of the VP1s and a subsequent structural phylogeny for classifying *Marnaviridae* viruses by their hosts. These findings will be crucial for inferring the host-specific receptor-binding mechanism in *Marnaviridae* viruses.

## 1. Introduction

Viruses are prevalent in the oceans and impact marine microbial communities by regulating the mortality of marine microorganisms such as diatoms [1]. Some groups of marine algae viruses have been isolated in the termination phase of algal blooms and caused cell lysis, which could be a reason for the sudden disappearance of algal blooms [2,3]. One group of these marine algal viruses is the picorna-like *Marnaviridae* viruses that display a narrow host range, which strongly engages with their host-specific mortality [4]. Evidence has been piled up that the strain-specific lysis of *Marnaviridae* virus-infecting eukaryotic phytoplankton is restructuring the phytoplankton communities such as algal blooms, which in turn play a critical role with regards to nutrient cycling, food web, and biomass in the aquatic ecosystem [4].

Abundant positive-sense, single-stranded ((+)ss)RNA picorna-like viruses have been discovered in aquatic samples, and their sequence diversity in RNA viromes has been highlighted [5,6,7]. These picorna-like viruses infect diverse marine unicellular algae [8,9,10,11,12,13] and currently belong to the virus family *Marnaviridae* according to the International Committee on Taxonomy of Viruses [14]. Similar to the icosahedral capsid of the *Picornaviridae* viruses, that of the *Marnaviridae* viruses adapts a common pseudo-T = 3 symmetry with a conserved jelly-roll structural fold [15]. The capsid shell of the *Marnaviridae* viruses is usually made up of three structural proteins (VP1–3) and a short VP4 peptide that is a self-cleaved product of the precursor of VP3 or VP2 (VP0) [14,15,16].

*Picornaviridae* viruses bind to host cell receptors via a deep canyon structure on VP1, described as a long-standing canyon hypothesis [17,18], and some of them could transmit via a non-canyon way using their prominent surface traits on the VP1 [19,20,21]. The genome-release conformational intermediates of the VP1 N-terminus and the egress formation at the 2-fold axes on the capsid are observed in non-infectious A-particles and empty particles and are linked to the canyon- or non-canyon-mediated transmission processes in the *Picornaviridae* viruses [22,23,24,25,26,27,28,29,30,31,32]. The VP4 peptides are released from the capsid and form a membrane pore for genome release during the transmission processes in the picornaviruses [33,34,35], whereas the Aichi virus lacks VP4 but presents a non-post-processed VP0 with a VP4-like pore-forming function using its N-terminal extension [36]. The empty virions display the dramatically expanded structure of the capsids that could be interpreted as another intermediate of a canonical uncoating process [37]. Hence, it is also critical to resolve the atomic model of the empty capsid to understand the entire transmission process. The transmission mechanism of the *Marnaviridae* viruses is less clear; however, the VP1 protein is thought to be responsible for receptor-binding since high amino acid sequence diversity has been identified in the VP1-encoding open-reading frame in *Marnaviridae* viruses, which may also reflect their host-specificity [6]. 

Previously, we have determined the first capsid structure of a *Marnaviridae* Chaetoceros tenuissimus RNA virus type II (CtenRNAVII) that infects *C. tenuissimus* [15]. The VP1 has a unique EF-Loop projected on the capsid surface, while it lacks canyons and pocket factor binding sites [15], and therefore the transmission mechanism of the CtenRNAVII does not meet previously reported mechanisms in picornaviruses. Our research interest also lies in how the host-specificity of the *Marnaviridae* viruses can be determined by their unique transmission mechanism. The structural diversity of the *Marnaviridae* viruses that infect different algae hosts will be a key to understanding their host-specific transmission. Thus, this paper reports on the structure of Chaetoceros socialis forma radians RNA virus 1 (CsfrRNAV) that infects *C. socialis* [11] as a distinct *Marnaviridae* virus.

Herein, we present both 3.0 Å full and 3.1 Å empty capsid structures of the CsfrRNAV. The first empty capsid structure provides new insights into genome packaging and transmission intermediates in *Marnaviridae* viruses. Together with the previously reported CtenRNAVII capsid structure, we have rigorously evaluated the structural features that could be critical for the common and specific transmission mechanisms for these viruses. For a future research vision, we also describe a potential use of AlphaFold2 structural prediction of the VP1 proteins and structure-based phylogeny for classifying them by host specificity.

## 2. Materials and Methods

### 2.1. Virus Propagation and Sample Purification

CsfrRNAV was produced as previously described [11]. Briefly, late exponentially growing cultures of *C. socialis f. radians*, strain NIES-3713, were inoculated with CsfrRNAV suspensions (1% *v*/*v*) and were grown for two weeks at 15 °C. The resultant lysate was passed through a 0.4-µm pore-size polycarbonate membrane filter (Nuclepore, Merck, Darmstadt, Germany) to remove cellular debris. Polyethylene glycol 6000 (Wako Pure Chemical Industries Ltd., Osaka, Japan) was added to the filtrate to achieve a final concentration of 10% (*w*/*v*), and the suspension was stored at 4 °C in the dark overnight. After centrifugation at 2600× *g* at 4 °C for 2 h, the pellet was washed with ultra-pure water and added with an equal volume of chloroform. After vigorous vortex mixing, the suspensions were centrifuged at 2600× *g* for 20 min at room temperature to remove the chloroform. Each water phase was collected and ultracentrifuged at 217,000× *g* for 4 h at 4 °C to collect the virus particles. The virus particles were resuspended in 300 µL of ultra-pure water (i.e., virus suspension). The crude virus samples were then loaded on a 5–50% (*w*/*v*) continuous sucrose gradient and ultracentrifuged at 57,522× *g* at 4 °C for 18 h. The virus fractions were pooled, diluted in TNE buffer (50 mM Tris, pH 7.4, 100 mM NaCl, and 0.1 mM EDTA), and ultracentrifuged at 139,190× *g* at 4 °C for 3 h. The virus pellet was resuspended in 100 µL TNE buffer and concentrated to 20 µL. 

### 2.2. Cryo-EM Data Collection and Single Particle Analysis

Three microliters of the CsfrRNAV sample were applied on a Quantifoil R2/2 copper grid and blotted for 2 s at 4 °C under 100% humidity using a plunge-freezing method (ThermoFisher, Vitrobot Mark IV). In total, 10,310 movie frames (40 frames/movie) were collected by a 300 kV Titan Krios cryo-EM microscope (Thermo Fisher Scientific, Waltham, MA, USA) equipped with a K3 summit direct electron detector (Gatan) and energy filtering. The summary of data processing is shown in Appendix A. Image processing and single particle analysis were performed using cryoSPARC 3.3.2 [38]. The movies’ frames, apart from the first and the last three frames, were motion corrected using a patch motion correction option. Contrast transfer function (CTF) values were estimated using a patch CTF correction option with a defocus range of −0.1 to −4.0 µm. In total 34,499 empty and 3984 full particles were finally selected by template picking and 2D classification. The cryo-EM model of the full particle was reconstructed by imposing an icosahedral (I) symmetry and to 3.4 Å with an FCS cutoff of 0.143 after an amplitude correction and tight masking. The local CTF values were further refined per particle, and then the final resolution was improved to be 3.0 Å (FCS cutoff = 0.143) for manual atomic modeling. An initial cryo-EM map of the empty capsid was reconstructed at 3.7 Å resolution and using 32,017 particles; however, due to high structural variability, the obtained map appeared smeared. To obtain a map suitable for model building, the initial particle dataset was first duplicated 60 times using the symmetry expansion job. The new particle set of 1,921,020 particles was used to perform a 3D variability analysis [39] using three orthogonal principal modes (Appendix A) and a resolution filter of 6.0 Å. The heterogeneity of the dataset was displayed as twenty clusters (Appendix A), of which only one produced a 3D reconstruction with a recognizable secondary structure. For the final reconstruction, the duplicate particles within that cluster were first removed, which resulted in 1302 particles. The following non-uniform refinement with imposed icosahedral symmetry resulted in reconstruction with a resolution of 3.2 Å, which was further improved to 3.1 Å in another non-uniform refinement conducted after global and local CTF refinement. 

### 2.3. Atomic Modeling and Refinement 

The atomic model of the VP1–3 capsid proteins were manually built in the full particle model using Coot 1.0.06-pre [40]. The initial model was refined using the ‘real-space refinement’ of Phenix-1.20.1-4487 [41]. The improved model was iteratively corrected and refined in Coot and Phenix. The final atomic model of VP1–3 was fitted in the empty particle model and refined. The validation statistics of the obtained atomic models (modeled in full and empty maps) are shown in Appendix A. 

### 2.4. Structural Analysis 

The obtained cryo-EM maps and atomic models were visualized using open-source PyMOL 1.4, UCSF Chimera, and ChimeraX [42,43]. Root-Mean-Square Deviation (RMSD) per residue calculation was performed by the local_rms command distributed by the extended PyMOL script collection (PSICO) module and the alignment of the module is based on TM-align [44].

### 2.5. Sequence- and Structure-Based Phylogeny 

The 18 *Marnaviridae* viruses currently classified by the ICTV that have complete capsid-coding regions were used for the phylogenetic analysis (Appendix A). The structures of two viruses, CsfrRNAV and CtenRNAVII, have been experimentally determined. For the remaining 16 viruses, the VP1 structures were predicted using AlphaFold 2.0 [45]. The sequences submitted to AlphaFold were selected based on a multiple sequence alignment using Clustal Omega [46] which also included the viruses with experimentally predicted structures as a reference. Appendix A includes the GenBank accession numbers, the sequences used for AlphaFold predictions, and the obtained pLDDT scores. The superimposition of the AlphaFold-predicted models is shown in Appendix A. AlphaFold2 is a newly developed accurate structure prediction software using database and neuronal networks and has widely been used for predicting functional structures [45,47]. For generating a structure phylogeny, all predicted and experimental structures of the VP1s were aligned using a multiple structural alignment algorithm (MUSTANG) [48], and all-vs-all overall RMSD values were calculated without the N-terminal arm region. The structure phylogeny was generated using the obtained RMSD values to be used for a distant matrix as previously described [15,49]. To generate a sequence phylogeny, the amino acid sequences of the VP1s of *Marnaviridae* viruses were aligned using MUSCLE in the MEGA11 software program [50]. The neighbor-joining phylogenetic tree was generated using the bootstrap method based on 1000 replicates, and the bootstrap values were calculated.

## 3. Results and Discussion

### 3.1. The Overall Capsid Structure of CsfrRNAV and Its Capsid Proteins

The CsfrRNAV capsid structures were determined using cryo-electron microscopy (cryo-EM) with 3984 full and 1302 empty particles. The atomic models of the CsfrRNAV full and empty capsids at final resolutions of 3.0 Å and 3.1 Å (Fourier shell correlation, FSC = 0.143) were built manually based on the cryo-EM 3D reconstruction map (Appendix A). The capsid shell was built in a pseudo-T = 3 symmetry and was composed of three VPs from VP1–3 (Figure 1A,B). In total, 785 and 718 amino acid residues were modeled for full and empty capsid proteins according to the 3D maps, respectively. For CsfrRNAV full particles, residues A34–L275 were modeled for VP2, C332–F609 were modeled for VP3, and S625–L894 were modeled for VP1 (Figure 1C). The first 33 amino acid residues (M1–P33), T239–S243, S276–R331, and S610–A624, and the last two residues D895–F896, were not visible and were unassigned (Figure 1C). VP1s surround the 5-fold axes and form plateaus on the surface of the CsfrRNAV capsid. Every two VP2s form a dimer, and together with VP3, each dimer of VP2 and VP3 builds a 3-fold (Figure 1B). Eight β-strands (βC_1_, D_2_, E_1_, E_2_, F, G, H, I) were observed in VP1 and formed two β-sheets (Figure 1B,C). In total, 12 β-strands (βX_1_, X_4_, B, C, D_1_, D_2_, E, F_1_, F_2_, H, I_1_, I_2_) composing two β-sheets in VP2 and 12 β-strands (βB, C_1_, C_2_, C_3_, D_1_, D_2_, E_1_, E_2_, G, H, I_1_, I_2_) composing two β-sheets in VP3 were found (Figure 1B,C). Residues A34–G87 and S625–S637 are invisible and unmodeled in the empty CsfrRNAV map in comparison with the full one.

All VPs are made of rich α-helices and β-sheets structures. Most β-strands/sheets are structurally conserved compared with those in CtenRNAVII VPs. The domain swapping is observed in the N-terminal extension of the VP2 (Figure 1B) as previously described in a *Marnaviridae* virus and primordial *Picornaviridae* viruses [15,21,51]. The prominent EF-Loop and canyon-obstructing CD-Loop are presented in the CsfrRNAV capsid (Figure 1D), which are similar to those of the CtenRNAVII capsid [15]. Between VP4 and VP3, there is a self-cleavage site GR/SRP compared with the site GY/SRP reported previously (Appendix A) [15]. CsfrRNAV VP1 has a DDF motif (residues 876–878), which is similar to the EDF motif in CtenRNAVII (Appendix A) [15]. Unlike CtenRNAVII VP2 which has a putative EDV cleavage motif, CsfrRNAV VP2 has a DNV motif (residues 255–257), instead (Appendix A) [15]. The cleavage motif in CsfrRNAV VP3 is possibly PSF (residues 585–587), while the one in CtenRNAVII is DDF (Appendix A) [15]. Unlike CtenRNAVII, which has VP1–4, the CsfrRNAV capsid lacks VP4 (Figure 1B,C) [15]. The N-terminal domain of the CsfrRNAV VP3 does not have an extension of the Aichi virus VP0-like non-cleaved VP4 region (Figure 1C). The amino acid sequence alignments of the CsfrRNAV and CtenRNAVII VP genes indicate that the CsfrRNAV lacks 35 amino acids near the VP2 C-terminal region (Appendix A, dashed line); however, the expected VP4 region remains in the gene sequence (Appendix A). The amino acid sequence of the VP4 region is present, but its structure is absent within the CsfrRNAV capsid. Thus, the VP4 is seemingly not an essential factor for virus transmission in CsfrRNAV. In spite of the VP4 absence, the overall capsid structure of CsfrRNAV displays structural features that are similar to those in the previously resolved CtenRNAVII capsid, implying that both *Marnaviridae* viruses employ common transmission mechanisms. However, local structural differences on their surface VPs should reflect the host-specific infectivity to different algae host *C. tenuissimus* or *C. socialis* species.

### 3.2. Empty Particle Structure and Variability

The empty capsid shows high conformational variabilities (Appendix A) compared with the full capsid (Appendix A). The full particles are more homogeneous and thus have fewer conformational variabilities (Appendix A). The 3D variability is observed as the rocking pentagonal structural units composed of five asymmetric units surrounding the 5-fold axis (5-fold structural units) in the empty capsids (Appendix A). The expansion of the 5-fold structural unit is also observed in the empty capsid (Appendix A). The 5-fold structural units were initially formed during the particle assembly [52,53] and were also partially removed from the capsid during the uncoating processes in picornaviruses [54,55,56]. The structural variability of the 5-fold structural units in empty capsids could reflect their structural plasticity and mobility in the procapsid and uncoating intermediates. Such rocking motions might be crucial for dynamic association or dissociation of the 5-fold structural units during capsid assembly and uncoating. 

The overall size of the empty CsfrRNAV particle (approximately 318 Å in diameter) is slightly larger than that of the full one (approximately 311 Å in diameter). The overall full and empty capsid features are similar, except that the 3-fold structures built with VP2s and VP3s are more densely and tightly assembled in the full capsid (Figure 2A,B). Clear gaps are observed at 2-fold axes in the empty particles (Figure 2A,B). The atomic models of each VP in the full and empty capsid are shown in Figure 2C. The large portion of the N-terminus in VP2 is apparently invisible in the empty capsid; however, the other VPs show no obvious structural differences (Figure 2C). The observed gaps seem to contribute to the rocking motions of the 5-fold structural units in the empty capsid and might also reflect its assembly and uncoating intermediates. However, unlike picornaviruses, no obvious conformational changes have been observed in the VP1 N-terminus between the full and empty capsids (Figure 2C). Along with the VP4 absence, this could also hint that a picorna-like transmission mechanism does not exist in CsfrRNAV. Empty capsids in *Picornaviridae* viruses have also been identified as reflecting procapsid intermediate formation naturally during particle assembly and genome packaging as a precursor of a mature virion [28,57]. The RNA-capsid interaction via W38 residue at the N-terminal part in VP2 is important for particle assembly in enterovirus [53], while other papers have described the VP3 contribution to the RNA interactions [32]. The VP2 N-terminal domain swapping could enhance particle stability in picornaviruses [51]. The VP2 N-terminal extension is visible only in the full capsid, indicating that this extension is probably important for particle stabilization, genome recruitment, and genome packaging. 

### 3.3. Local Structural Differences of VP1 in Two Marnaviridae Viruses

Local surface structural differences of VPs, especially VP1, will reflect the host-specific binding mechanism in *Marnaviridae* viruses. Hence, the structural diversity was thoroughly analyzed in the CtenRNAVII and CsfrRNAV VPs. The overall RMSD values were calculated between the two VP structures of CtenRNAVII and CsfrRNAV. The VP1s show the highest structural diversity with the RMSD value of 1.544 Å (164 Cα), 1.024 Å for VP3s (174 Cα), and 0.900 Å for VP2s (158 Cα). Apart from the N- and C-termini in VP1, the structural differences are mainly observed in E_1_E_2_ and CD loops and are much lower in the unique EF-Loop (Figure 3B,C). The E_1_E_2_-Loop shows the most structurally diverse segment (RMSD = 7–8 Å) (Figure 3B,C). The structural diversity of the E_1_E_2_ and CD loops seems to affect the structure of the connecting β-strand as well. VP2 does not show apparent structural differences apart from the N-terminal arm, while VP3 shows some structural differences in the surface loops adjacent to the VP1 E_1_E_2_-Loop (Appendix A). To acquire more ideas on the implications between the VP structures and host specificity, the VP1 structures from 16 registered *Marnaviridae* viruses were predicted with a high predicted Local Distance Difference Test score (pLDDT) (mostly 70–100) using AlphaFold2 (Appendix A). Apart from N-terminal domains, the predicted structures are reliable, which are also assessed by structural errors between the experimental and predicted CsfrRNAV VP1 atomic models (Appendix A). These predicted structures were aligned with those of CtenRNAVII and CsfrRNAV for structural comparisons (Figure 4A). Within the predicted VP1 structures, the E_1_E_2_-Loop shows the highest structural diversity (RMSD = 6–8 Å), and the CD and EF-loops show some diversities (RMSD = 4–6 Å) (Figure 4B).

*Marnaviridae* viruses present an extra EF-Loop unlike *Picornaviridae* viruses, which implies that they possess a unique receptor-binding mechanism [15]. The EF-Loop could be universally important for the transmission of *Marnaviridae* viruses; however, it is not the fully determining factor for their host specificity. The E_1_E_2_ and/or CD loops probably play a critical role in their host-specific binding mechanism, such as functioning as a binding site for a specific receptor in each algal host. To understand the dynamics of the algal bloom caused by each algae species, it is necessary to invent an efficient high-throughput method, the structural phylogenetic tree, for classifying *Marnaviridae* viruses by reflecting their algal hosts. In our recent studies [49,58], the structure-based phylogenetic tree of viral capsid proteins improves classifying distantly related viruses. Hence, structural phylogeny is generated using the aligned structures of VP1s (Figure 4D) and is then compared with the sequence phylogeny (Figure 4C). Two phylogenetic trees display clear differences in cladding the *Marnaviridae* viruses. The structural phylogeny of VP1s will reflect their local structural diversity, which links to the host-specific receptor-binding in *Marnaviridae* viruses and can be applied for a better prediction of their targeted algal hosts. It is interesting to perform further structural predictions and analysis of the *Marnaviridae* viruses in metatranscriptomics RNA virome data and to evaluate the reclassified phylogeny.

## Figures and Tables

**Figure 1 viruses-14-02369-f001:**
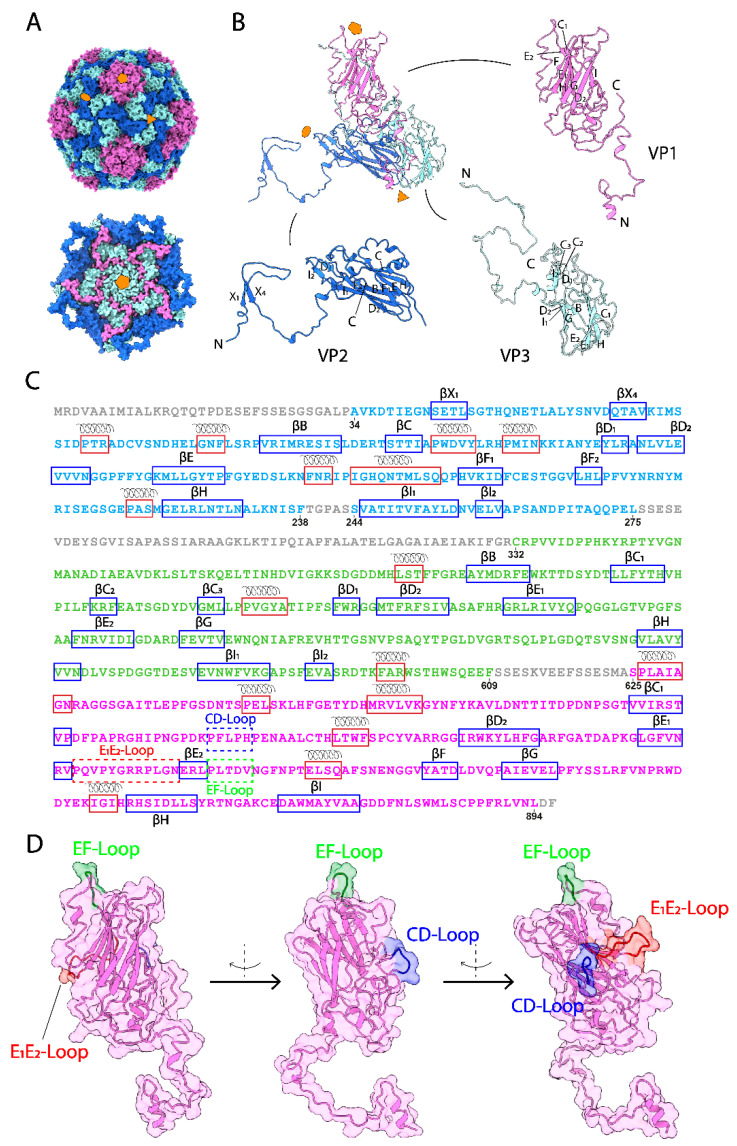
Overall CsfrRNAV CP structure and VPs atomic models. (**A**) Overall geometry of the CsfrRNAV CP (**top**) and the CP 5-fold interior view (**bottom**). Pentagon, triangle, and ellipse refer to 5-fold, three-fold, and two-fold, respectively. (**B**) Atomic models of VP1–3. N or C terminus is labeled. Other letters refer to each of the β-strands in each VP, according to the information shown in (**C**). (**C**) Amino acid sequence organization of the CsfrRNAV CP. Residues for VP1 are in pink, residues for VP2 are in blue, and residues for VP3 are in green. Red and blue rectangular boxes display the α-helices and β-strands. Dash boxes show special CD-Loop, E_1_E_2_-Loop, and EF-Loop in VP1. Unmodeled residues are in gray. (**D**) Images of three special loops in VP1. The CD-Loop, E_1_E_2_-Loop, and EF-Loop are shown in blue, red, and green, respectively, with the rotation to the left.

**Figure 2 viruses-14-02369-f002:**
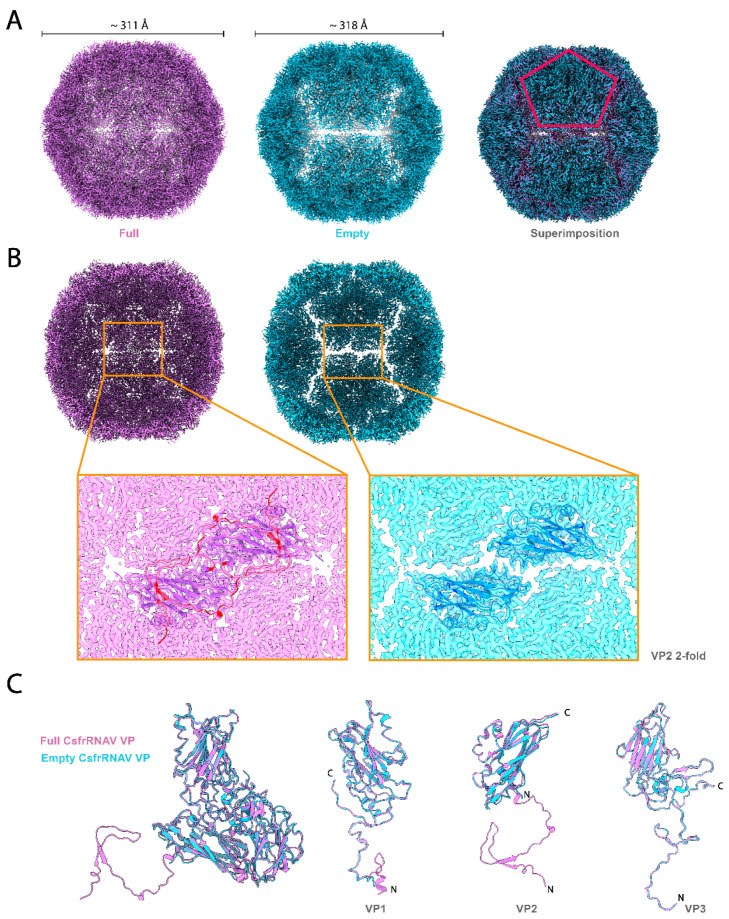
Capsids and the VPs superimposition of the full and empty CsfrRNAV. (**A**) Map views of the CPs of the full (purple) and empty (blue) particles. The superimposition is shown on the right. The red pentagon refers to one of the VP 5-folds. Both particle diameters of the full and empty 3D reconstructed maps are measured in ChimeraX and are approximately 311 Å and 318 Å, respectively. (**B**) Back halves of the full (purple) and empty (blue) CsfrRNAV models and the close-up views of the gap between 5-folds. VP2 structures are shown in purple and blue for full and empty particles, respectively. VP2 N-terminal extension/swap in full particles is colored red. (**C**) VP structure comparisons between full and empty particles.

**Figure 3 viruses-14-02369-f003:**
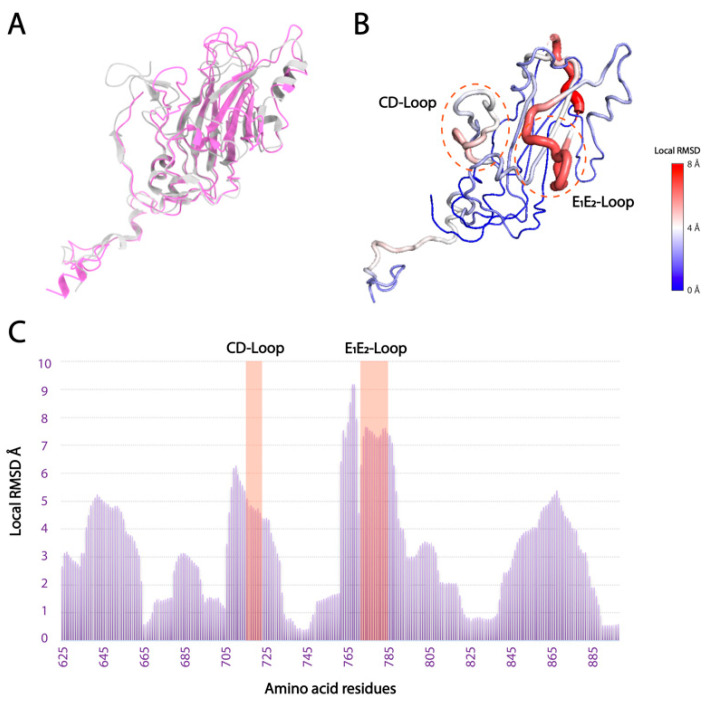
Local structural differences between CtenRNAV and CsfrRNAV VP1s. (**A**) Superimposition of CtenRNAV (silver) and CsfrRNAV (purple) VP1s. (**B**) The circled regions display low-to-high conformational differences (from blue to red). (**C**) The overall plot for presenting RMSD value per residue. Unique loops (CD and E_1_E_2_-Loops) are highlighted in red.

**Figure 4 viruses-14-02369-f004:**
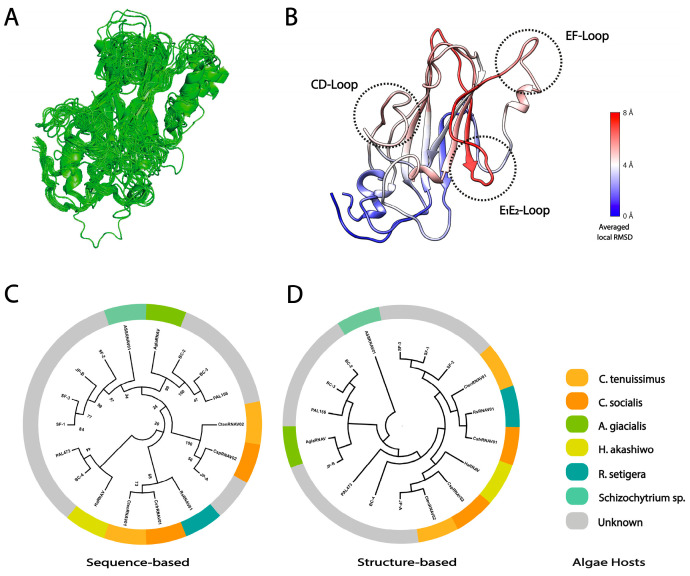
Predicted structural variation in *Marnaviridae* viruses VP1 and the structure phylogeny. (**A**) Pairwise alignment of 16 predicted structures of *Marnaviridae* viruses VP1s and two experimental structures of CtenRNAVII and CsfrRNAV VP1s. The N-terminal arms were trimmed for the alignment due to their low accuracy in the predicted structures (Appendix A); they are not involved in the transmission mechanism. (**B**) Averaged RMSD per residue map of the *Marnaviridae* virus VP1s calculated against CsfrRNAV VP1. The residues are colored by the attribute of the averaged residue RMSD on the CsfrRNAV VP1 atomic model. (**C**) Amino acid sequence-based phylogeny of the *Marnaviridae* virus VP1s. (**D**) Structure-based phylogeny generated by the predicted and experimental *Marnaviridae* virus VP1 structures. (**C**,**D**) Each clade is colored by identified algae hosts.

## Data Availability

The full and empty cryo-EM models of the CsfrRNAV capsid were deposited in the EMDB repository, entries EMD-15823 and EMD-15830. The atomic models of the VP1–3 in full and empty particles were deposited in the PDB repository, entries 8B38 and 8B3J.

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
