# Peer review of "Structural Insights into Common and Host-Specific Receptor-Binding Mechanisms in Algal Picorna-like Viruses"

_viruses, 2022, doi:10.3390/v14112369_

Round 1

Reviewer 1 Report

In this manuscript, the authors reported the cryo-EM structures of the full and empty capsids of CsfrRNAV, which reveal the structural mechanisms of viral capsid assembly and uncoating. By comparing solved and predicted structures of VP1 among Marnaviridae viruses, different structure features were inentified that might contribute to the host tropism. The paper is well written and there are minor issues to address: 

1. Please provide a high-resolution image for Figure 1C and Figure 4C&D (or maybe increase the font size).

2. Does the structure-based phylogeny better clustered algae hosts that are more closely related than the sequence-based phylogeny?

Author Response

In this manuscript, the authors reported the cryo-EM structures of the full and empty capsids of CsfrRNAV, which reveal the structural mechanisms of viral capsid assembly and uncoating. By comparing solved and predicted structures of VP1 among Marnaviridae viruses, different structure features were identified that might contribute to the host tropism. The paper is well written and there are minor issues to address: 

Point 1: Please provide a high-resolution image for Figure 1C and Figure 4C&D (or maybe increase the font size).

Response: The submitted PDF version reduced the image resolution. The new high-resolution images are attached with increased quality.

Point 2: Does the structure-based phylogeny better clustered algae hosts that are more closely related than the sequence-based phylogeny?

Response: Thank you for the comment. The available information of the Marnaviridae viruses’ host is still limited to date, and thus we would avoid concluding whether we have obtained better clusters or not. However, through our further massive structural predictions of metagenomics virome data with a collaboration of bioinformaticians, this method will potentially help clustering the viruses by the host-specificity.

In our recently published studies in Munke et al., mBIO (2022), DOI: 10.1128/mbio.00156-22 and Wang et al., bioRxiv (2022), DOI: 10.1101/2022.09.29.510062, the structural phylogeny of capsid proteins helps reclassifying genetically distantly related viruses. The picorna-like Marnaviridae viruses are classified in one family, however it is known that they are less homologous due to their narrow host spectrum infecting diverse algae host. Thus, we are also expecting the importance of the method for future classification of the Marnaviridae viruses.

We added new sentences in the discussion section and referred these papers. (Page 8, Lines 306-307, 313-315)

Reviewer 2 Report

The Paper of “Structural insights into common and host-specific transmission mechanisms in algal picorna-like viruses” investigated the ommon and diverse structural features of the VP1 surface between the Marnaviridae viruses. They have also tested the potential usage of AlphaFold2 for structural prediction of the VP1s and a subsequent structural phylogeny for classifying Marnaviridae viruses by their hosts. These findings it is helpful to understand the host-specific transmission mechanism in Marnaviridae viruses.

This paper has some innovation and scientific values. The data are detailed and reliable, and the results can support the conclusions of the full text. It can be recommended for publication after minor modification. Some specific comments as follows:

1.     Delete the last sentence of the Abstract, becaues it is lack of directly evidence about it.

2.     It is necessary added some contents about this virus and it’s host, it can help the reader to better understand the ecological role of this virus in the algal bloom event.

3.     Improve the resolution of Figure S1.

4.     The references are irregular (such as the For example, italic, case, etc), it need to carefully to revise according to the Viruses’s style.

Author Response

The Paper of “Structural insights into common and host-specific transmission mechanisms in algal picorna-like viruses” investigated the common and diverse structural features of the VP1 surface between the Marnaviridae viruses. They have also tested the potential usage of AlphaFold2 for structural prediction of the VP1s and a subsequent structural phylogeny for classifying Marnaviridae viruses by their hosts. These findings it is helpful to understand the host-specific transmission mechanism in Marnaviridae viruses. 

This paper has some innovation and scientific values. The data are detailed and reliable, and the results can support the conclusions of the full text. It can be recommended for publication after minor modification. Some specific comments as follows:

Point 1: Delete the last sentence of the Abstract, because it is lack of directly evidence about it.

Response: We have deleted the sentence starting from “which is decisive...” from the abstract since it is still speculative.       

Point 2: It is necessary added some contents about this virus and it’s host, it can help the reader to better understand the ecological role of this virus in the algal bloom event.

Response: Thank you for the comment. The general importance of algae viruses for bloom events are described in Introduction section (Page 1, Lines 27-32) and referred from two papers [2], [3]. The reference [4] in the next sentence summarizes the ecological role of the picorna-like Marnaviridae viruses. Thus, we have extended the description and added one more sentence to address the comment with the reference. (Page 1, Lines 33-36)

Point 3: Improve the resolution of Figure S1.

Response: The submitted PDF version reduced the image resolution. The Figure S1 is attached.

Point 4: The references are irregular (such as for example, italic, case, etc), it needs to carefully to revise according to the Viruses’s style.

Response: The reference format has been revised.

Reviewer 3 Report

The manuscript by Wang et al. describes a structural study of the algal virus Chaetoceros socialis forma radians RNA virus V (CsfrRNAV), infecting a unicellular algae C. socialis f. radians. The structure of the intact virus particles and of empty capsids was resolved through cryo-EM, atomic modeling, and structural analysis. This is an interesting project that presents novel and significant findings about mechanisms of binding and cell entry of marnaviruses into their algal hosts. Technologies and approaches used are adequate to the proposed objectives. The manuscript is generally well-written and may need minor-to-modest editing. One thing that stands out is the confusing use of the term ‘transmission’ for apparently two separate processes of the early stages of the infection cycle, i.e. binding of the virus to the host receptor and entry/release of the virus genome into the host cell. These should be carefully separated through proper editing of the text. Hence, the term ‘transmission’ in the title is confusing and must be replaced with something like ‘binding’. Transmission as a term cannot be really applied to a system of a unicellular host and a virus, it is all about receptor binding and release of the virus genome into the cell.

The manuscript is recommended for publication once the terminology use is corrected and harmonized.   

Specific points to address:

l. 14 – binding or receptor-binding, not ‘transmission’

l. 23 – similarly, replace ‘transmission’ with ‘binding’

l. 45 – consider replacing ‘transmit’ with ‘bind’

Author Response

The manuscript by Wang et al. describes a structural study of the algal virus Chaetoceros socialis forma radians RNA virus V (CsfrRNAV), infecting a unicellular algae C. socialis f. radian. The structure of the intact virus particles and of empty capsids was resolved through cryo-EM, atomic modeling, and structural analysis. This is an interesting project that presents novel and significant findings about mechanisms of binding and cell entry of marnaviruses into their algal hosts. Technologies and approaches used are adequate to the proposed objectives. The manuscript is generally well-written and may need minor-to-modest editing. One thing that stands out is the confusing use of the term ‘transmission’ for apparently two separate processes of the early stages of the infection cycle, i.e. binding of the virus to the host receptor and entry/release of the virus genome into the host cell. These should be carefully separated through proper editing of the text. Hence, the term ‘transmission’ in the title is confusing and must be replaced with something like ‘binding’. Transmission as a term cannot be really applied to a system of a unicellular host and a virus, it is all about receptor binding and release of the virus genome into the cell.

The manuscript is recommended for publication once the terminology use is corrected and harmonized.   

Point 1: Specific points to address: 

  1. 14 – binding or receptor-binding, not ‘transmission’
  2. 23 – similarly, replace ‘transmission’ with ‘binding’
  3. 45 – consider replacing ‘transmit’ with ‘bind’

Response: As you suggested, the transmission should have several processes. A large part of our results and discussions are focused on the receptor binding step. We have thoroughly checked the manuscript including your specific points and replaced the word “transmission” to “receptor binding” or similar terms.